# One-Step Microwell Plate-Based Spectrofluorimetric Assay for Direct Determination of Statins in Bulk Forms and Pharmaceutical Formulations: A Green Eco-Friendly and High-Throughput Analytical Approach

**DOI:** 10.3390/molecules28062808

**Published:** 2023-03-20

**Authors:** Ibrahim A. Darwish, Hany W. Darwish, Nourah Z. Alzoman, Awadh M. Ali

**Affiliations:** Department of Pharmaceutical Chemistry, College of Pharmacy, King Saud University, P.O. Box 2457, Riyadh 11451, Saudi Arabia

**Keywords:** statins, spectrofluorimetry, microwell assay, green analytical approach, high-throughput analysis

## Abstract

This study describes the development of a one-step microwell spectrofluorimetric assay (MW-SFA) with high sensitivity and throughput for the determination of four statins in their pharmaceutical and formulations (tablets). These statins were pitavastatin (PIT), fluvastatin (FLU), rosuvastatin (ROS) and atorvastatin (ATO). The MW-SFA involves the measurement of the native fluorescence of the statin aqueous solutions. The assay was conducted in white opaque 96-microwell plates, and the fluorescence intensities of the solutions were measured by using a fluorescence microplate reader. The optimum conditions of the assay were established; under which, linear relationships with good correlation coefficients (0.9991–0.9996) were found between the fluorescence intensity and the concentration of the statin drug in a range of 0.2–200 µg mL^–1^ with limits of detection in a range of 0.1–4.1 µg mL^–1^. The proposed MW-SFA showed high precision, as the values of the relative standard deviations did not exceed 2.5%. The accuracy of the assay was proven by recovery studies, as the recovery values were 99.5–101.4% (±1.4–2.1%). The assay was applied to the determination of the investigated statins in their tablets. The results were statistically compared with those obtained by a reference method and the results proved to have comparable accuracy and precision of both methods, as evidenced by the t- and F-tests, respectively. The green and eco-friendly feature of the proposed assay was assessed by four different metric tools, and all the results proved that the assay meets the requirements of green and eco-friendly analytical approaches. In addition, ever-increasing miniaturization as handling of large numbers of micro-volume samples simultaneously in the proposed assay gave it a high-throughput feature. Therefore, the assay is a valuable tool for the rapid routine application in the pharmaceutical quality control units for the determination of statins.

## 1. Introduction

Atherosclerosis is the principal cardiovascular disease that causes morbidity and mortality worldwide [1]. Elevated plasma level of total cholesterol (hypercholesterolemia) is predictive of atherosclerosis [2]. Lowering cholesterol levels can arrest or reverse atherosclerosis and significantly decrease the morbidity and mortality rates associated with atherosclerosis [3]. The most efficient and widely prescribed drugs for the treatment of hypercholesterolemia are the 3-hydroxy-3-methylglutaryl coenzyme A (HMG-CoA) reductase inhibitors (more commonly known as statins) [4]. Statins control the elevated levels of low-density lipoprotein (LDL) cholesterol, a well-established risk factor for atherosclerosis [5]. Statins specifically inhibit HMG-CoA reductase, the enzyme that catalyzes the conversion of HMG-CoA to mevalonate, the rate-limiting step in the de novo biosynthesis of cholesterol in the body (Figure 1).

The reduction of HMG-CoA to mevalonate is regulated by nicotinamide adenosine di-phosphate cofactor (NADPH), and its activity is functioning under a feedback mechanism [6]. All the approved HMG-CoA reductase inhibitors (statins) share a common pharmacophore, which is an essential intermediate in cholesterol biosynthesis (Figure 1A); however, these drugs have distinct pharmacokinetic and pharmacodynamic properties that markedly affect their overall efficacy, safety, and potential non-LDL actions. There is strong evidence of the therapeutic benefits of the use of statin in the reduction of morbidity and/or mortality due to cardiovascular via reducing the levels of LDL-C. Also, statins play an important role in the regression or stabilization of coronary atheromatous plaques. The survival benefits of using statin in therapy are independent of the level of the cholesterol baseline. Statins vary significantly in their rate of absorption, degree of protein binding, the extent of excretion, metabolism, hydrophilicity, and interactions with other drugs. These differences are attributed to their chemical structures, which affect their efficacy, the spectrum of action, tolerance and interactions [5].

The effective and safe therapy with statins is mainly dependent on the quality of their pharmaceutical dosage forms in terms of the active drug content and content uniformity. For quality control of statins, a proper analytical technique is required. The existing techniques for the determination of stains in their pharmaceutical dosage forms are mostly liquid chromatography, electrochemical techniques and spectrometry; these techniques have been reviewed in many reports [7,8,9,10,11,12,13]. In the pharmaceutical industry, spectrofluorimetric assays are the most convenient and widely used. They are characterized by their inherent simplicity, ease of use, low cost, and broad availability in the majority of pharmaceutical quality control laboratories, high sensitivity and selectivity [14,15]. Different spectrofluorimetric assays were developed for the determination of statins in their bulk and/or dosage forms [16,17,18]. Unfortunately, all these assays involved the conventional practice which involved the non-automated analysis using the volumetric flasks/cuvettes. This practice has a limited throughput and consumes large volumes of expensive and toxic organic solvents [19]. Therefore, these assays do not meet the requirements of green and high-throughput analytical approaches used in the pharmaceutical industry [20,21,22,23,24].

The aim of the study was the development of a new microwell-spectrofluorimetric assay (MW-SFA) for the determination of statins. The proposed MW-SFA was developed and validated for four statins; their chemical structures are given in Figure 1. The chemical names, molecular formulae, molecular weights, and appearance of the investigated statins are summarized in Table 1, and their comparative clinical efficacies along with pharmacokinetics are given in Table 2. The proposed assay was conducted in white opaque 96-microwell assay plates for analysis and the native fluorescence intensities of the investigated statins were measured by using a fluorescence microplate reader. The proposed assay meets the needs of pharmaceutical industries and their quality control units in terms of ease of analysis, consumption of small volume samples, safety to the health and environment, production of non-hazardous waste, and high analytical throughput. The proposed assay overcomes the drawbacks of the existing spectrofluorometric assays for statins.

## 2. Results and Discussion

### 2.1. Design and Strategy of Assay Development

Previous reports [7,8,9,10,11,12] and the spectra, generated in our laboratory, confirmed that statins involved in this study have native fluorescence (Figure 1). This native fluorescence of statins is expected because their chemical structures contain fluorophoric moieties such as extended conjugations and rigid fused aromatic rings with fluorescence-promoting substituents [25]. Therefore, the development of a spectrofluorimetric assay for statins was considered in this study. All the reported spectrofluorimetric assays for statins involved the traditional manual practice which uses volumetric flasks/cuvettes in the analysis. Therefore, these assays have a limited throughput that does not meet the needs of quality control laboratories in pharmaceutical industries [20,21,22,23,24]. In addition, these assays consume large volumes of samples in costly organic solvents, and more importantly, the analysts are exposed to the harmful adverse health hazards of the organic solvents [19]. Furthermore, these assays were developed for the individuals of the investigated statins; the development of only one assay procedure for all statins would be very convenient. For these reasons, this study was directed to develop an alternative assay with higher throughput, uses smaller volumes of samples, eco-friendly/green assay, and can be used for all the statins. Recently, our laboratory successfully employed a fluorescence microplate reader in the development of microwell-based spectrofluorimetric assays for the quantitation of the active drug contents in their pharmaceutical dosage forms [26,27]. These assays are considered as a green analytical approach and provided high throughput analysis. To date, no spectrofluorimetric assay has been developed for the statins using a fluorescence microplate reader to achieve better sensitivity in a short analysis time. For these reasons, the present study has committed to the development and validation of this methodology for statins.

### 2.2. Development of MW-SFA

#### 2.2.1. Fluorescence Spectral Characteristics

The fluorescence spectra (excitation and emission) of all statins were recorded in methanol (Figure 2). The spectra showed that the maximum excitation wavelengths (λ_excitation_) of statins were 335, 305, 315 and 277 nm for PIT, FLU, ROS and ATO, respectively; whereas the maximum emission wavelengths (λ_emission_) were 405, 395, 365 and 390 nm PIT, FLU, ROS and ATO, respectively. Initially, the measured intensities of the native fluorescence of all statins were encouraging for employment in the development of a spectrofluorimetric assay with high sensitivity for their determination in their pharmaceutical formulations (tablets). The following sections describe the experiments involved in the optimization of the assay variables and discuss the results.

#### 2.2.2. Optimization of Assay Variables

It is well-established that the native fluorescence of any fluorophoric molecule can be affected by its physical and/or chemical surrounding environments, such as the solvent, organic medium and pH of the solution. Therefore, experiments were conducted to establish the most appropriate variables for the proposed MW-SFA of the statins. These experiments were carried out in white opaque microwell plates because they show minimal background signals and are devoid of any crosstalk between the wells.

##### Effect of Solvent

The effect of solvent on the fluorescence intensity of statins was studied using different solvents with varying polarities (water, acetonitrile, ethanol, methanol, isopropanol, and dichloromethane). The polarity indices of these solvents were 10, 5.7, 5.2, 5.1, 3.9 and 3.5, respectively [28]. It was found that higher fluorescence intensities were achieved when solvents with higher polarity indices were used (water, acetonitrile, methanol and ethanol). Lower fluorescence intensities were achieved when a less polar solvent (dichloromethane) was used (Figure 3). This solvent polarity-dependent fluorescence intensity was related to the relatively different abilities of statins to interact with different solvents. Also, statin molecules have non-symmetric dipoles and exhibit varying rates of relaxation in solvents with varying polarities. Furthermore, the intermolecular arrangement of the fluorophoric moieties in the excited states of the statin molecules may lead to the formation of hydrogen bonding and cause temporal effects on the fluorescence of statin molecules [29]. Among the solvents which gave higher fluorescence intensities, water was selected as a solvent in all subsequent experiments because it is not expensive and more importantly is health-safe and fulfills the green approach in the pharmaceutical analysis.

##### Effect of Surfactant

In spectrofluorimetric analysis, it has been documented that adding a surfactant or macromolecule to the natively fluorescent drug solution may enhance its fluorescence intensity and consequently increase the assay sensitivity [30]. Therefore, the effect of different surfactants of different categories on the fluorescence intensities of statins was studied. These surfactants were anionic (sodium lauryl sulfate: SLS), cationic (dodecyl dimethyl ammonium bromide: DDAB), non-ionic (tween 20) and macromolecule (carboxymethyl cellulose: CMC). The results indicated that tween 20 had an enhancing effect (~17%) on the fluorescence intensity of all the cited statins, as compared with their aqueous solutions containing no surfactant (Figure 4). Also, CMC had a quenching effect (~30%) on the fluorescence intensities of the statin-aqueous solutions. Although the achieved enhancement effect of tween 20 on the fluorescence intensity of statins, we preferred to rule it out from further experiments. This decision was taken to establish an assay with a simple and green procedure for statins, taking into account that the original intrinsic sensitivity without any surfactant is satisfactory for the determination of statins in their dosage forms, and ever-still comparable with that of the reported spectrofluorimetric assays for statins [16,17,18]. In addition, the use of water only without any organic additives to the analysis media adds the advantage of the green and eco-friendly feature to the proposed assay.

##### Effect of pH

The effect of pH on the fluorescence intensity of statin solutions (1 µg mL^–1^) was assessed by measuring the fluorescence intensity of buffered solution with varying pH (2–12). The results revealed that the pH in the range of 2–9 had no effect on the fluorescence intensity, and higher pH values (9–12) had a negative effect. The results are shown for PIT (Figure 5); whereas the same behavior was observed for the other statins (FLU, ROS and ATO). Previous studies [31,32] revealed that statins undergo pH-dependent interconversion between their hydroxy acid and lactone forms. Also, it was shown that for both forms interactions can affect the lipid-lowering activity of the drug at the molecular level. This transformation did not occur in the fluorophoric moieties of the statin molecule; therefore, their fluorescence intensities were not affected by changing the pH over the range of 2–9. The decrease in the fluorescence intensity at a highly alkaline media (at pH > 9) was attributed to the possible degradation of the molecules. This assumption was supported by a previous study involving the degradation susceptibility of statins in highly alkaline media [33]. According to these findings, all the subsequent experiments were carried out without using a buffer solution. This added advantages to the procedures such as less cost, eco-friendly, and health safety.

##### Effect of Sample Solution Volume

To select the most appropriate volume of statin solution which is dispensed in each well of the assay plate, different volumes (50, 100, 150 and 200 µL/well) of PIT solution, as a representative example, were dispensed. The fluorescence intensities of these volumes were measured, and relative standard deviation (RSD) values of the readings were calculated. The results demonstrated that the fluorescence intensity increased, in linear proportion, with the volume of the drug solution (Figure 6). Also, the precision of the readings, expressed as their RSD values, decreased as the volume of the solution increased, and the lowest RSD value (1.9%) was obtained when 200 µL/well was used. Accordingly, 200 µL/well was used in all the subsequent experiments.

### 2.3. Validation of MW-SFA

The MW-SFA was validated according to the guidelines of the International Council of Harmonization (ICH) for validation of analytical procedure [34]. Validation was conducted in terms of linearity, sensitivity, precision, accuracy, ruggedness and robustness.

#### 2.3.1. Linearity and Sensitivity

The calibration graphs for the determination of the stains by the proposed MW-SFA were constructed by plotting the fluorescence intensity as a function of the corresponding concentrations of each statin drug (Figure 7). The regression analysis for the results was conducted. The linear equations with their correlation coefficients and other statistical parameters were computed. A summary of these statistical parameters is given in Table 3.

The limits of detection (LOD) and limits of quantification (LOQ) of the MW-SFA were calculated. LOD, defined as the minimum detectable statin drug concentration, was calculated for each drug using the formula: 3.3 × standard deviation of intercept/slope. LOQ, defined as the statin drug concentration that can be quantified with satisfactory accuracy and precision under the stated procedures of the MW-SFA, was calculated using the formula: 10× standard deviation of intercept/slope. The calculated LOD and LOQ values are given in Table 4. It is wise to note that the sensitivity of the proposed assay, in terms of its LOD and LOQ, is higher than that of the reported spectrofluorimetric assays for stains involving their native fluorescence.

#### 2.3.2. Precision and Accuracy

The intra- and inter- proposed MW-SFA for statins assay precisions of the proposed MW-SFA were evaluated by replicate analysis of nominated concentrations of each statin solution. For the intra–assay precision, the samples (*n* = 3) were analyzed as a batch; however, for the inter-assay precision, the samples were analyzed on two consecutive days. In both cases, RSD values were calculated and used as a measure of precision. The recovery percentages, relative to nominated concentrations were calculated and taken as a measure for accuracy. It was found that the RSD values were in the ranges of 1.34–2.10 and 1.65–2.50% for the intra– and inter-assay precision, respectively (Table 4). These low RSD values revealed the high precisions of the proposed MW-SFA for the determination of the statins. This high precision was attributed to the simplicity of the assay procedures which involve the measuring of the native fluorescence of the statins without any extra experimental manipulations (e.g., derivatization reaction) which may negatively affect the precision of the readings. In addition, the precise dispensing of the solutions into the microwells by the 8-channel pipettes is ultimately reflecting on the high of the proposed MW-SFA.

The results of the recovery study at both intra- and inter-assay accuracy are given in Table 4. The recovery values were in the range of 98.5–101.4% (±1.2–1.6) and 98.6–101.8% (±1.5–2.4) for intra- and inter-assay accuracy, respectively. These good recovery values confirmed the accuracy of the proposed MW-SFA for the determination of statins.

#### 2.3.3. Robustness and Ruggedness

Principally, the robustness of any assay is defined as its ability to maintain the reliability of its analytical results upon minor changes in its experimental conditions. The proposed MW-SFA involves the direct measuring of the native fluorescence of the investigated statins in their aqueous solutions without any experimental manipulations; therefore, it is considered a robust assay.

The ruggedness of the proposed MW-SFA, defined as its ability to reproduce the results in different circumstances without the occurrence of unexpected differences in the obtained results, was evaluated in terms of analyst-to-analyst and day-to-day reproducibility. The RSD values for the analysis of statin samples by two different analysts on two different days did not exceed 2.4%, respectively. This result confirmed the ruggedness of the proposed MW-SFA for statins.

### 2.4. Application of MW-SFA and Statistical Comparison with a Reference Method

The aforementioned results of the validation study of the proposed MW-SFA indicated its applicability to the determination of the cited statins in their pharmaceutical formulations (tablets). The samples of the commercial tablets were subjected to analysis by the MW-SFA for their contents of the corresponding statins; the results are given in Table 5. The label claims percentages of the labeled drug content were found in the range of 99.4–101.4% (±0.5–1.6). Subsequently, in a separate set of experiments, the same tablets were analyzed by a pre-validated reported reference method [35]. The label claim percentages were found in the range of 99.6–100.9% (±1.2–1.9). The results obtained by the proposed MW-SFA were statistically compared with those obtained by the reference method in terms of accuracy and precision by t- and F-tests, respectively. The results of t- and F-tests revealed that there were no significant differences between both methods at a *p*-value of <0.05, at a 95% confidence level (Table 5).

### 2.5. Evaluation of the Eco-Friendship and Greenness of MW-SFA

The proposed MW-SFA was evaluated in terms of its eco-friendship and greenness to determine its impact on the health and environment. In general, the microwell-based assays employing a microplate reader for measuring the signals mostly meet the requirements of green analytical chemistry (GAC) principles because the miniaturization of analytical techniques usually reduces the volume of samples, reagents, and production of wastes, compared to conventional techniques. For precise evaluation of this general view, the greenness of the proposed assay was evaluated by using the National Environmental Method Index, NEMI [36], Eco-Scale [37], Analytical Greenness, AGREE [38] and Green Analytical Procedure Index, GAPI [39] metric tools which provide precise and easy comprehensive assessments of the greenness of analytical procedures.

The NEMI tool was initially used for qualitative assessment of the greenness of the proposed MW-SFA. The NEMI’s circular pictogram encompasses four fields reflecting four different criteria of the analytical procedures: persistence, bio-accumulation potential, toxicity (PBT), hazardous chemicals, corrosiveness, and waste. When the assay meets the required criterion of each field, it takes a green color. Upon applying these criteria to the MW-SFA, 3 quadrants in the pictograms took a green color (Figure 8) because they fulfilled their requirements by the tool. The fourth quadrant corresponding to the hazardous did not take a green color because of the use of methanol to prepare the stock solutions of statins, and a very small amount was used per sample.

In the Eco-Scale tool, a numerical score was given to each parameter of the proposed MW-SFA. The score of the assay obtained by the Eco-Scale is the result of the subtraction of the total penalty points (PPs) of the test assay from 100 points (score of an ideal green analysis). The PPs of the MW-SFA were assigned to each parameter of the Eco-Scale assessment tool (type and amount of chemical reagents, energy consumed by the instrument, occupational hazards, and production/treatment of waste). Subsequently, the MW-SFA’s score was ranked on the Eco-Scale where the assay is considered as excellently green when its score is higher than 75, acceptable when its score is higher than 50, and inadequate when its score is less than 50 [37]. The calculation demonstrated that the proposed MW-SFA got a total PPs of 9 and earned a score of 91 (Table 6) which reveals the eco-friendship of the assay to merit its safe application for quality control purposes in pharmaceutical industries.

Regarding the AGREE tool [38], it is a technique for identifying the environmental and occupational dangers involved in the analytical procedure. It provides a clock-shaped pictogram with a circumference split into 12 pieces, based on the 12 principles of GAC. On a color scale, each item is handled as a separate parameter. The center of the AGREE chart indicates the overall acceptance color and assessment score on a scale of 0 to 1. The proposed MW-SFA assay achieved an AGREE score of 0.83 of 1 (Figure 8). This high score confirms the greenness of the assay.

The most recent tool, GAPI [39] was also applied to assess the greenness of the proposed MW-SFA. The tool classifies the greenness of each stage of an analytical procedure, using a color scale, with three levels of evaluation. In GAPI, a specific symbol with five pictograms is used for quantitative assessment; from green through yellow to red, which are corresponding to the low, medium and high impact of each step of the methodology on the environment. Each field reflects a different aspect of the test analytical procedure and the field is filled green when the aspect meets certain requirements. The visual-based presentation of the GAPI’s pictogram enables the researchers to make their own judgments about conflicting green criteria. As shown in the GAPI’s pictogram of the proposed MW-SFA (Figure 8), it is mostly shaded with green and yellow colors for most of the assessment criteria, indicating their low impact on the environment.

Conclusively, the overall assessment of the eco-friendly and greenness of the proposed MW-SFA described herein meets the requirements for green analytical approaches for routine use for quality control purposes in pharmaceutical industries.

### 2.6. Advantages of MW-SFA over the Reported Spectrofluorimetric Assays

All the reported spectrofluorimetric assays for the statins involve 10 mL volumetric flasks/cuvettes and a conventional spectrofluorometer for measurements of the signals. A summary of the characteristics of these assays is given in Table 7. Compared with these assays, the proposed MW-SFA is superior to all these assays as it has the following advantages. In the proposed MW-SFA, the sample solutions were dispensed simultaneously by the multi-channel pipettes into the 96–microwell plates (200 μL sample volume) instead of using 10 mL volume methanol in conducting the conventional reported spectrofluorimetric assays. The simultaneous dispensing of the sample by the multi-channel pipettes was time-saving. The assay uses water as a solvent, rather than organic solvents in the reported assays, which allows health-safe sample handling and non-hazardous waste production. The assay procedures relied on a one-step direct measuring of the native fluorescence of the statins by the plate reader; this implies the simplicity of the assay. The use of a plate reader enables the simultaneous reading of 96 wells of the plate in a few seconds; this feature enables automation and high-throughput analysis. Also, the use of a plate reader is advantageous as it is considered an energy-saving instrument as it requires a low amount of energy (<0.1 kWh per sample as per the Eco-Scale and AGREE metric tools for assessing the greenness of analytical procedures. Furthermore, the proposed assay offers comparable high sensitivity and better linearity than the reported assays. Overall, the proposed MW-SFA is valuable for use in routine work and quality control purposes in pharmaceutical industries.

## 3. Experimental

### 3.1. Apparatus

Fluorimeter (FP-8200: JASCO Co. Ltd., Kyoto, Japan), with 1 cm quartz cells was used for the recording of the fluorescence spectra. The slit width of both the excitation and emission monochromators was set at 5 nm. The instrument is operated and controlled by Spectra Manager^®^ software provided with the instrument. Fluorescence microplate reader (FLx800: Bio-Tek Instruments Inc., Winooski, USA) empowered by KC junior software, provided with the instrument. Microprocessor laboratory pH meter (BT-500: Boeco, Hamburg, Germany). Purelab Flex water purification system (ELGA Veolia Ltd., High Wycombe, UK).

### 3.2. Standards, Pharmaceutical Formulations and Materials

Standard materials of statins were obtained from AK Scientific Inc. (Union City, California, USA). Their purities were >99% and were used as received. The pharmaceutical formulations (tablets) of the investigated statins used in this study are given in Table 8. Corning^®^ 96–well white microplates with flat bottoms were purchased from Merck & Co., Inc. (Rahway, New Jersey, NJ, USA). Finnpipette™ adjustable single and 8-channel pipettes were obtained from Thermo Fisher Scientific Inc. (Waltham, MA, USA). BRAND^®^ PP reagent reservoirs with lids for dispensing the solutions by the multichannel pipettes were purchased from Merck KGaA (Darmstadt, Germany). Britton-Robinson buffer solution was prepared at a pH range of 2–12. Sodium lauryl dodecyl, tween 20, dodecyl dimethyl ammonium bromide and carboxymethyl cellulose were purchased from Sigma–Aldrich Chemicals Co. (St. Louis, MO, USA). All solvents were of spectroscopic grade (Merck, Darmstadt, Germany). All other chemicals used throughout the work were of analytical grade. Distilled-deionized water was obtained from the Purelab Flex water purification system.

### 3.3. Preparation of Solutions

#### 3.3.1. Standard Solutions

An accurately weighed amount (25 mg) of each statin drug was transferred into a 25–mL volumetric flask and dissolved in ~15 mL of methanol. The volumes were completed to the mark with the same solvent to provide stock standard solutions (1 mg mL^–1^). The stock solutions were kept at 4 °C until use. The working standard solutions were prepared by dilution of the stock solutions with water, in 10-mL volumetric flasks, to give final concentrations of 1, 2, 5 and 50 µg mL^–1^ for PIT, FLU, ROS and ATO, respectively. Appropriate aliquots of methanol were used in each solution to compensate for the differences in methanol percentages and keep it constant in all solutions.

#### 3.3.2. Tablets Sample Solutions

Ten tablets of each tablet’s brand were weighed and crushed to a fine powder. An accurately weighed quantity of the fine powder equivalent to 25 mg of the active ingredient (statin drug) was transferred into a 25-mL calibrated flask, dissolved in ~15 mL methanol. The contents of the flask were swirled well for 5 min to ensure complete dissolution of the active drugs and then completed to the volume with methanol. The mixtures were mixed well, filtered, and the first portions of the filtrates were rejected. Aliquots (2 mL) of the filtrates were diluted quantitatively with water to yield solutions with nominated concentrations in the linear range of each particular statin drug.

### 3.4. Procedure of MW-SFA

Aliquots (200 μL) of the standard or sample solution containing drug concentration in the range of 0.2–5, 0.5–10, 2–25 and 10–200 µg mL^–1^ for PIT, FLU, ROS and ATO, respectively, were separately transferred into each well of a 96-microwell plate. The fluorescence intensities were measured by using a fluorescence microplate reader. Wells containing 200 μL of water were used as blank and the measured fluorescence intensities of the blank wells were subtracted from those of the sample wells. The fluorescence intensities were measured at 405, 395, 365 and 390 nm (λ_emission_) for PIT, FLU, ROS and ATO, respectively, after excitation at 335, 305, 315 and 277 nm (λ_excitation_) for PIT, FLU, ROS, and ATO, respectively.

## 4. Conclusions

This study described, for the first time, the development of an MW–SFA for the estimation of the statins in their bulk form and tablets. The proposed assay meets the demands of quality control laboratories in pharmaceutical industries in terms of ease of analysis, use of small volume samples, health/environment safety, non-hazardous waste production, and high analytical throughput as it enables the analyst to quickly process hundreds of samples and collect massive data as a batch, and consequently saves resources in terms of time, effort and reagent quantity. The assay broadens the horizon for the efficient use of microplate readers for rapid quantitative assays of active pharmaceutical ingredients and manifests the suitability for application in pharmaceutical quality control units. Moreover, the cumulative results of the study described herein confirmed the higher sensitivity of the approach for measuring the native fluorescence with detection limits as low as 0.1–4.1 µg mL^−1^. Overall, the proposed assay was validated according to ICH guidelines for the routine analysis of the studied statins with acceptable precision and accuracy, confirming that the proposed assay is a potential alternative approach to replace the existing assays and overcome their major drawbacks.

## Figures and Tables

**Figure 1 molecules-28-02808-f001:**
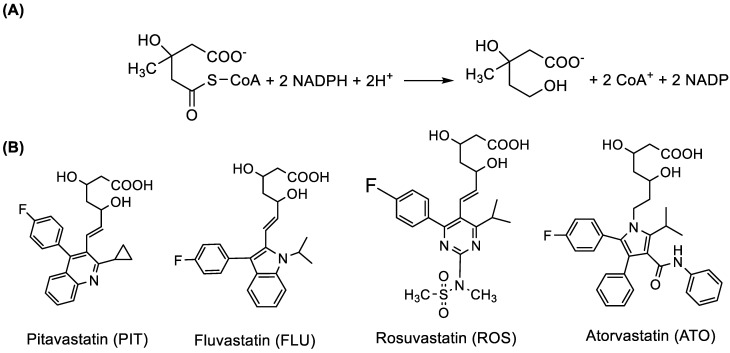
The HMG-CoA reductase reaction (**A**) and the chemical structures of the investigated HMG-CoA reductase inhibitors (statins) with their abbreviations (**B**).

**Figure 2 molecules-28-02808-f002:**
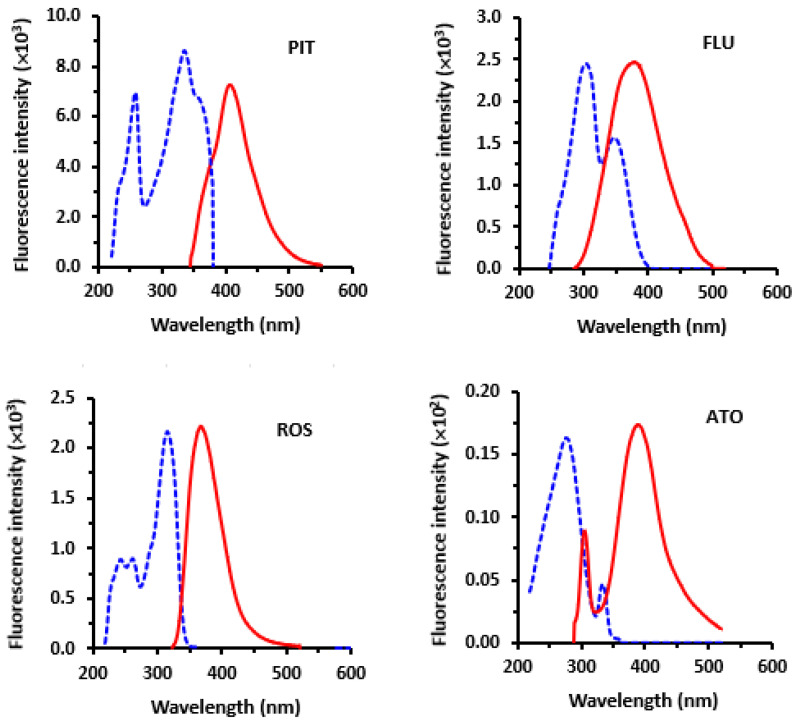
Excitation (-----) and emission (_____) spectra of the investigated statin solutions (10 µg mL^–1^). PIT, FLU, ROS and ATO are the abbreviations of pitavastatin, fluvastatin, rosuvastatin and atorvastatin, respectively.

**Figure 3 molecules-28-02808-f003:**
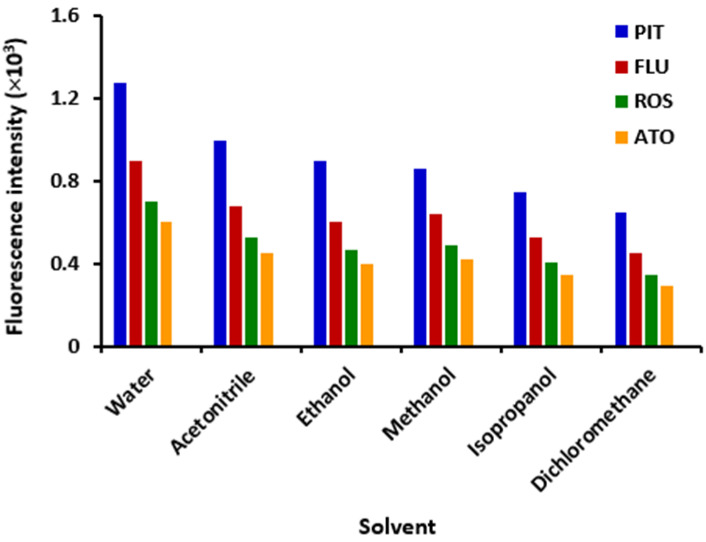
Effect of solvent on the fluorescence intensity of statin solution (in water). Concentrations were 1, 2, 5 and 50 µg mL^–1^ for PIT, FLU, ROS and ATO, respectively. Values are the mean of 3 determinations.

**Figure 4 molecules-28-02808-f004:**
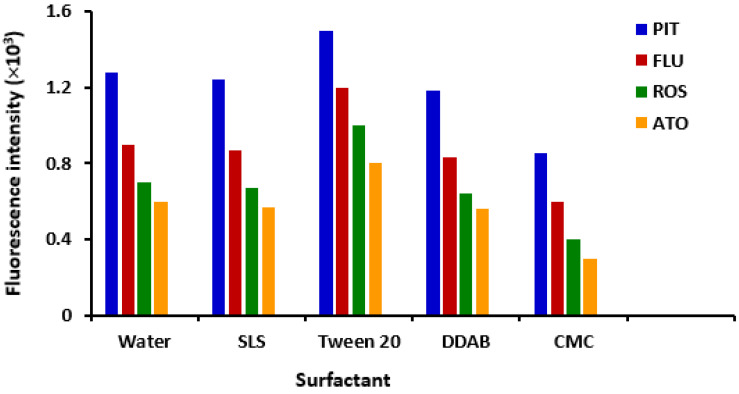
Effect of type of surfactant on the fluorescence intensity of statin solution (200 ng mL^–1^) compared with an aqueous solution without any surfactant. Concentrations were 1, 2, 5 and 50 µg mL^−1^ for PIT, FLU, ROS and ATO, respectively. Values are the mean of 3 determinations. The abbreviations (SDS, DDAB and CMC) denote sodium lauryl sulfate, dodecyl dimethyl ammonium bromide and carboxymethyl cellulose, respectively.

**Figure 5 molecules-28-02808-f005:**
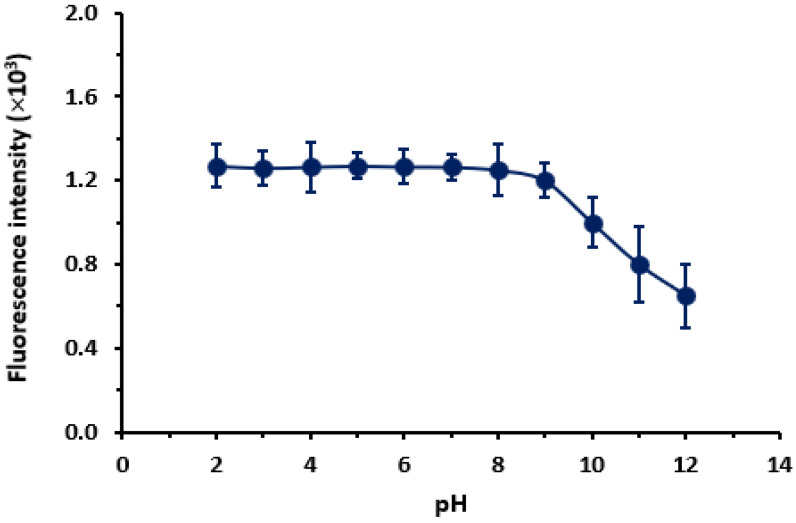
Effect of pH on the fluorescence intensity of PIT solution (1 µg mL^–1^). Values are mean of 3 determinations ± SD.

**Figure 6 molecules-28-02808-f006:**
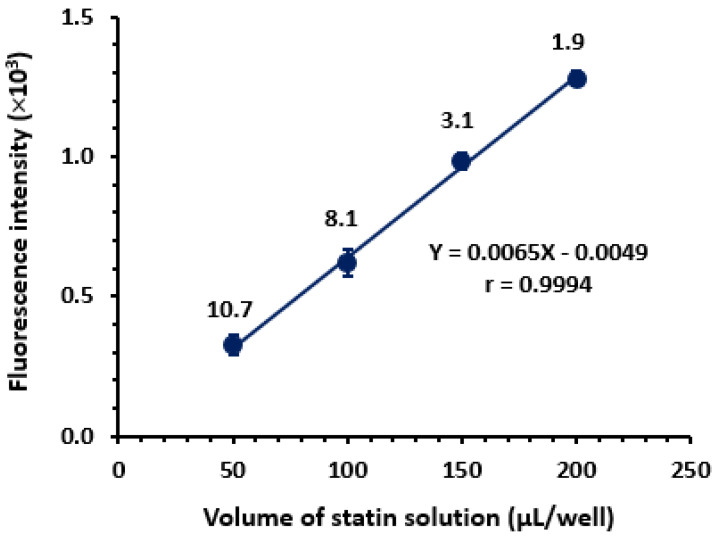
Effect of volume of PIT solution (1 µg mL^–1^) on the fluorescence intensity measured by the fluorescence microwell reader. The linear fitting equation of fluorescence intensity (Y-axis) vs. volume (X-axis) and its correlation coefficient (r) is given on the graph. The values are the mean of 3 determinations ± SD and the figure given on each point is the relative standard deviation of the readings.

**Figure 7 molecules-28-02808-f007:**
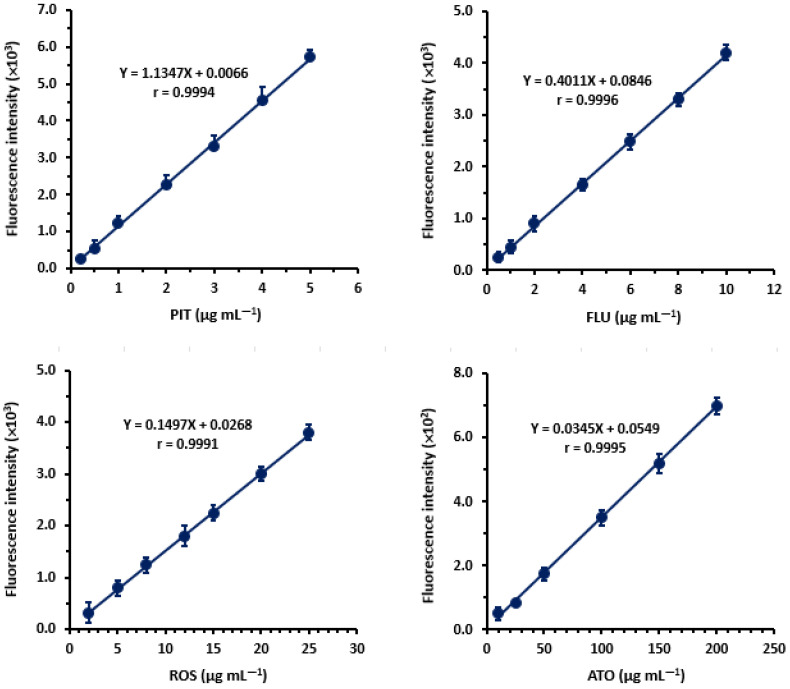
Calibration curves for determination of statins by the proposed MW-SFA. Values of fluorescence intensity are mean of 5 determinations ± SD.

**Figure 8 molecules-28-02808-f008:**
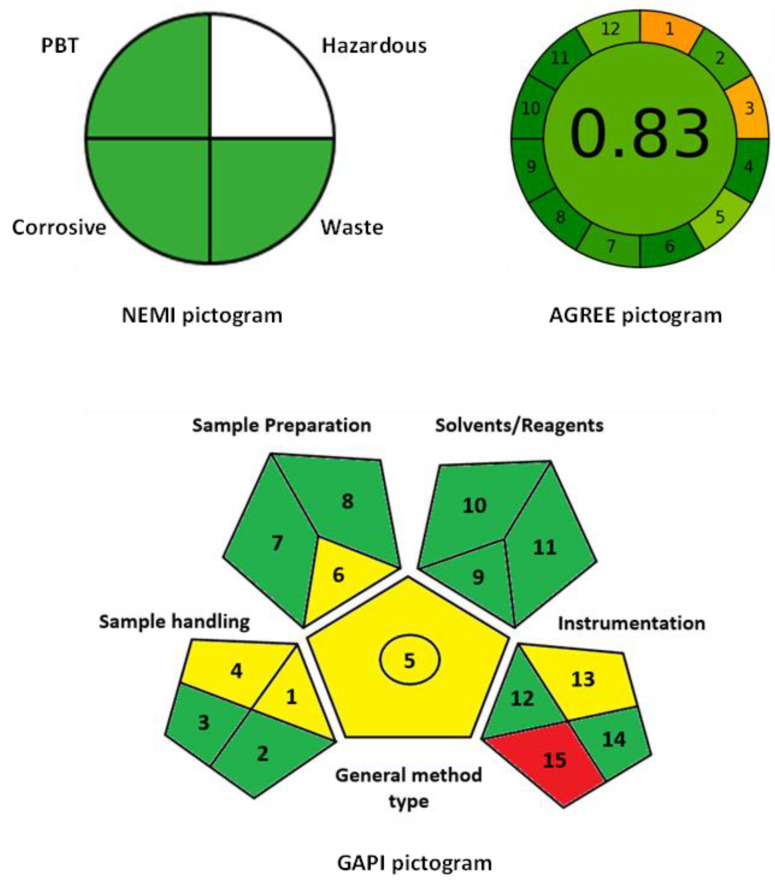
Results of NEMI, AGREE and GAPI metric tools for evaluation of the greenness of the proposed MW-SFA for statins.

**Table 1 molecules-28-02808-t001:** The names, chemical nomenclatures, and physical properties of the investigated statins.

Drug Name (Abbreviation)	Chemical Nomenclature	Molecular Formula	Molecular Weight	Appearance
Pitavastatin (PIT)	(E,3R,5S)-7-(2-cyclopropyl-4-(4-fluorophenyl) quinolin-3-yl)-3,5-dihydroxyhept-6-enoic acid.	C_23_H_24_FNO_4_	421.46	White to yellowish-white powder
Fluvastatin (FLU)	(3R,5S,6E)-7-[3-(4-Fluorophenyl)-1-(propan-2-yl)-1H-indol-2-yl]-3,5-dihydroxyhept-6-enoic acid	C_24_H_26_FNO_4_	411.47	Yellowish-white powder
Rosuvastatin (ROS)	(3R,5S,6E)-7-[4-(4-Fluorophenyl)-6-(1-methylethyl]-3,5-dihydroxy-6-heptenoic acid	C_22_H_28_FN_3_O_6_S	481.54	White powder
Atorvastatin (ATO)	(3R,5R)-7-[2-(4-fluorophenyl)-3-phenyl-4-(phenylcarbamoyl)-5-propan-2-ylpyrrol-1-yl]-3,5-dihydroxyheptanoic acid	C_33_H_35_FN_2_O_5_	558.65	White to off-white crystalline powder

**Table 2 molecules-28-02808-t002:** Comparative clinical efficacies and pharmacokinetics and of the investigated statins ^a^.

Parameter ^b^	Pitavastatin (PIT)	Fluvastatin (FLU)	Rosuvastatin (ROS)	Atorvastatin (ATO)
IC_50_ (nM)	6.8	27.6	5.4	8.2
Bioavailability (%)	51	19–29	20	12
Protein binding (%)	>99	>99	88	80–90
T_max_ (h)	~0.5	0.5–1	3	2–3
C_max_ (ng/mL)	67	448	37	27–66
t_1/2_ (h)	~1.4	0.5–2.3	20.8	15–30
Metabolites	Inactive	Inactive	Active	Active
Urinary excretion (%)	15	6	10	negligible
Fecal excretion (%)	79	90	90	Major route
Reduction in LDL (%)	44	22–36	45–63	26–60

^a^ Values are given based on an oral dose of 4 mg for PIT and 40 mg for the other drugs. ^b^ T_max_ = time to maximum plasma concentration; C_max_ = maximum plasma concentration; t_1/2_ = elimination half-life; IC_50_ = concentration to produce a 50% reduction in the activity of statins.

**Table 3 molecules-28-02808-t003:** The optimum conditions of the proposed MW-SFA for statins and its analytical parameters.

Conditions	PIT	FLU	ROS	ATO
λ_excitation_ (nm)	258	305	315	275
λ_emission_ (nm)	405	380	370	305
Linear range (µg mL^–1^)	0.2–5	0.5–10	2–25	10–200
Intercept (au)	0.0066 × 10^3^	0.0846 × 10^3^	0.02268 × 10^3^	0.0549 × 10^2^
Slope (au µg mL^–1^)	1.1347 × 10^3^	0.4011 × 10^3^	0.1497 × 10^3^	0.0345 × 10^2^
Correlation coefficient, r	0.9994	0.9996	0.9991	0.9995
Limit of detection (LOD, µg mL^–1^)	0.1	0.4	0.8	4.1
Limit of quantitation (LOQ, µg mL^–1^)	0.4	0.6	2.4	12.5

**Table 4 molecules-28-02808-t004:** Precision and accuracy of the proposed MW-SFA for determination of statins.

Statin	Relative Standard Deviation (RSD, %) ^a^				Recovery (% ± SD) ^a^	
	Intra-Assay, *n* = 3		Inter-Assay, *n* = 3		Intra-Assay, *n* = 3	Inter-Assay, *n* = 3
PIT	1.52		1.91		100.2 ± 1.4	98.6 ± 1.8
FLU	1.65		2.50		101.4 ± 1.2	100.2 ± 1.5
ROS	2.1		1.82		99.8 ± 1.5	101.8 ± 2.2
ATO	1.34		1.65		99.5 ± 1.6	99.7 ± 2.4

^a^ Values are the mean of three determinations.

**Table 5 molecules-28-02808-t005:** Analysis of statin-containing tablets and statistical comparison between the accuracy and precision of the proposed MW-SFA with the reference methods.

Tablets ^a^	Label Claim (% ± SD) ^b^			t-Value	*p*-Value	F-Value
	Proposed MW-SFA	Reference Method ^c^				
Levalo	101.3 ± 1.2	100.2 ± 1.5		0.9918	0.3774	0.6400
Levazo	100.1 ± 1.5	99.6 ± 1.4		1.2662	0.2742	1.1480
Lescol XL	99.7 ± 0.5	100.1 ± 1.1		0.5734	0.5971	0.2066
Crestor	99.4 ± 0.7	99.8 ± 1.2		0.4987	0.6442	0.3403
Ivarin	101.4 ± 1.1	100.9 ± 1.2		0.5320	0.6229	0.8403
Lipitor	100.3 ± 0.6	100.7 ± 1.3		0.4839	0.6538	0.2130
Atorva	101.0 ± 1.4	99.8 ± 1.9		0.8807	0.4282	0.5429
Atorlip	99.6 ± 1.6	100.2 ± 1.8		0.4315	0.6883	0.7901

^a^ The active ingredient and strength of each tablet formulation are given in Table 8. ^b^ Values are the mean of three determinations.^c^ Reference method: [35].

**Table 6 molecules-28-02808-t006:** Analytical Eco-Scale for assessing of the greenness of the proposed MW-SFA.

Eco-Scale Score Parameters	Penalty Points (PPs)
Reagents/word sign/no. of pictograms	
Distilled water/–/0	0
Methanol/danger/3	6
	Σ = 6
Instrument: Energy used (kWh per sample)	
Fluorescence microplate reader	0
pH meter	0
Vortex mixer	0
Sonicator	0
Centrifuge	0
	Σ = 0
Occupational hazardous	
Analytical process hermetic	0
Emission of vapors and gases to the air	0
	Σ = 0
Waste	
Production (<1 mL (g) per sample)	0
Treatment (No treatment involved)	3
	Σ = 3
Total PPs	9
Eco-Scale score	91 (100 − 9)

**Table 7 molecules-28-02808-t007:** Analytical characteristics of the reported spectrofluorimetric assays for determination of statins.

Statin	Solvent	Linear Range (μg mL^–1^)	LOD (μg mL^–1^)	LOQ (μg mL^–1^)	Application	Ref.
PIT	Methanol	1–30	0.01	0.10	Bulk drug and tablets	[18]
FLU	Ethanol	1–10	0.09	0.26	Bulk drug and tablets	[19]
ROS	Methanol	0.5–100	0.6	0.5	Bulk drug and tablets	[18]
ATO	Methanol	0.5–3	0.01	0.13	Tablets	[20]
	Acetic acid (5%, *v/v*)	1–4	0.01	0.13	Tablets	[20]

**Table 8 molecules-28-02808-t008:** The pharmaceutical tablets of the investigated statins.

Brand Name (Tablets)	Manufacturer	Manufacturer Address	Active Ingredient (Statin Drug)	Label Claim (mg/tablet)
Levalo	Kowa Pharmaceuticals America Inc.	Montgomery, AL, USA	PIT	4
Levazo	Algorithm Inc.	Zouk Mosbeh, Lebanon	PIT	2
Lescol XL	Novartis AG	Cambridge, MA, USA	FLU	80
Crestor	AstraZeneca plc.	Cambridge, United Kingdom	ROS	20
Ivarin	Tabuk Pharmaceuticals	Riyadh, Saudi Arabia	ROS	20
Lipitor	Pfizer Inc.	New York, NY, USA	ATO	20
Atorva	Jazeera Pharmaceutical Industries	Riyadh, Saudi Arabia	ATO	20
Atorlip	Globalpharma Co. L.L.C.	Dubai Arab Emirates	ATO	10

## Data Availability

Not applicable.

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
