# Peer review of "One-Step Microwell Plate-Based Spectrofluorimetric Assay for Direct Determination of Statins in Bulk Forms and Pharmaceutical Formulations: A Green Eco-Friendly and High-Throughput Analytical Approach"

_molecules, 2023, doi:10.3390/molecules28062808_

Round 1

Reviewer 1 Report

Please see my main suggestions:

Figure 1 is blurred. Pleased provide a better quality one. Do not save it in any format only print screen it from the original image to keep the fidelity.

L53-59 Role and importance of statins must be better detailed in the Introduction section. I suggest checking and referring to https://doi.org/10.3390/diagnostics10070483

L64. After each method, the proper reference must be added, not as a group of 7!!! References [7-13].

L68. 2 consecutive references must be cited as 16,17.

L70. involved. L71. involves. Please reshape avoiding repetition.

Aim of the study must be the last paragraph of the Introduction, highlighting the novelty/special aspects your paper brings to the filed. In the actual shape, there is a mix of tables, what wanted the author to do and no relevant aspect about you really aimed. Please reshape/restructure.

L92. Why 1-cm and not 1 cm?

According to the Instructions for authors, In the entire manuscript please revise Abbreviations as Acronyms/Abbreviations/Initialisms should be defined the first time they appear in each of three sections: the Abstract; the Main text; under the first Figure or Table. When defined for the first time, the acronym/abbreviation/initialism should be added in parentheses after the written-out form. Begin with the abbreviations used in the Tables 3-5 etc., and check the entire manuscript in this regard.

References should be provided in the MDPI style, with all the requested info in the Instructions for authors. Please revise.

Author Response

Please, find the enclosed file

Reviewer 2 Report

Please, see the attached document.

Author Response

Please, find the enclosed file
